# Cofactors of earlier uptake of modern postpartum family planning methods in Kenya

Nancy M. Ngumbau[1]*, Damaris Kimonge[1], Julia C. Dettinger[2], Felix Abuna[1], Ben Odhiambo[1], Laurén Gómez[3], Anjuli D. Wagner[2], Mary M. Marwa[1], Salphine Watoyi[1], Emmaculate Nzove[1], Jillian Pintye[4], Jared M. Baeten[2,5], John Kinuthia[1], Grace John-Stewart[2,3,5,6], Cyrus Mugo[1]

1 Research & Programs, Kenyatta National Hospital, Nairobi, Kenya, 2 Department of Global Health, University of Washington, Seattle, Washington, United States of America, 3 Department of Epidemiology, University of Washington, Seattle, Washington, United States of America, 4 Biobehavioral Nursing & Health Informatics, University of Washington, Seattle, Washington, United States of America, 5 Department of Medicine, University of Washington, Seattle, Washington, United States of America, 6 Departments of Pediatrics, University of Washington, Seattle, Washington, United States of America

* nancym390@gmail.com

## Abstract

There are limited data on uptake of postpartum family planning (FP), particularly in high HIV prevalence settings. We assessed the timing of modern postpartum FP initiation and the cofactors of earlier uptake using longitudinal data from a clinical trial conducted in Kenya to assess two models of PrEP delivery among pregnant and postpartum women (NCT#03070600). Time to uptake of modern postpartum FP was estimated using survival analysis methods, and Cox proportional hazard models were used to determine cofactors of earlier uptake of modern postpartum FP. Among 4,191 women, median age was 24 years, 17% were aged 15–19 years, 88% were in a steady relationship, 50% intended to be pregnant and 75% were multigravida. The median time to resumption of sex was 8 weeks postpartum versus 24 weeks for uptake of postpartum FP. At 6 weeks postpartum, 42% of women had resumed sex, versus 12% who took up FP; at 14 weeks, 79% versus 38%; at 6 months, 88% versus 59%; and at 9 months, 91% versus 80%, respectively. Injectables and implants were the most common FP methods. Approximately 3.3% of all women became pregnant during the 9-month postpartum period. Being older and having ≤4 children was associated with earlier uptake of modern postpartum FP. Women with lower education, primigravida, low social support, history of miscarriage/stillbirth, without a partner at enrolment, not residing with their partners, not receiving financial support from their partner and whose youngest child at enrolment was < 2 years had later uptake of postpartum FP. Women who were ambivalent about their immediate previous pregnancy took up postpartum FP later than those who intended to be pregnant. Our findings underscore the importance of addressing the individual, interpersonal, social and obstetric factors associated with timeliness of postpartum FP uptake during the

**Data availability statement:** The data that support the findings of this study are publicly available on GitHub - https://github.com/Ngumbau-Nancy/Cofactors-of-FP-uptake

**Funding:** The authors have declared that no competing interests exist. JMB is employed by Gilead Sciences, outside of this work. There are no patents or products related to this submission. The authors have no financial, non-financial, professional, or personal competing interests that may alter the adherence to PLOS ONE policies on sharing data and materials.

**Competing interests:** The authors have declared that no competing interests exist. JMB is employed by Gilead Sciences, outside of this work.

development and delivery of postpartum FP interventions, particularly in high HIV prevalence settings.

## Introduction

Despite the introduction of several family planning (FP) methods and the implementation of many FP programs worldwide, the global vision of meeting the FP needs of women of reproductive ages 15 – 49 years as defined by the Sustainable Development Goal (SGD) 3.7.1 is far from being reached [1]. Globally, the proportion of women whose FP needs are satisfied with modern methods (hormonal/artificial methods or minor surgery that interferes with reproduction allowing people to have sexual intercourse at any time with diminished risk of pregnancy [2]) is 77% [3] and more than 218 million women in low and middle-income countries (LMICs) still experience unmet need for FP [4]. Although substantial gains in the use of FP methods have been witnessed in Sub-Saharan Africa, this region has the lowest proportion of use of modern FP methods among women who want to avoid pregnancy (56%) and is the only region in which injectables are the most prevalent method [3].

Since the inception of a national FP policy and program in 1967 in Kenya [5], modern FP use has increased remarkably from 7% in 1978 to 57% among married women. Additionally, 59% of sexually active unmarried women currently use modern FP [6]. Despite these gains, there are huge variations in the prevalence of modern FP use in the country ranging from 3% in Wajir county to 82% in Embu county impacting the ongoing efforts to increase the overall FP prevalence in the country [6]. Siaya and Homa Bay counties, which also have the highest HIV prevalence, are among the leading counties with a high burden of unmet need for family planning (27% and 17% respectively), high levels of women aged 15–19 years who have ever been pregnant (20.9% and 23.2% respectively) contributing to increased unintended pregnancies, maternal and child mortality rates and vertical transmission of HIV [6–8].

Postpartum period is a key time when women need FP to decrease the risk of unintended pregnancies and short birth intervals [9]. Evidence suggests that uptake of postpartum FP – any modern method of FP to prevent unintended pregnancies during the first 12 months after childbirth or pregnancy loss – can reduce maternal and childhood mortalities by 30% and 10% respectively [10,11], improve their mental well-being and allow women to attain their education and career goals thus fostering economic stability and the quality of life of their households and communities [12,13]. Most women wish to delay a subsequent pregnancy for at least 2 years, in line with WHO recommendations for safe pregnancy spacing [14]. However, the postpartum modern contraceptive prevalence rate in LMICs remains low [9,11,15]. In Kenya, the estimated unmet need for FP among postpartum women 23 months post-delivery is 57% [16]. During this period, approximately 80% of the women with live births in Kenya interact with the healthcare systems as they seek postnatal care and infant vaccinations, [6] presenting an opportunity to take up modern methods of FP [17,18]. However, women in the postpartum period have unique biological and physiological

considerations, such as amenorrhea post-delivery, and reduced or no sexual activity [9] which reduce their perceptions of pregnancy risk [19]. This, coupled with concerns about contraceptive side effects and safety concerns, partner disapproval, and low facility-based delivery of postpartum FP services results in decreased uptake of postpartum FP despite the need for these methods [20].

To boost the demand and utilization of postpartum family planning (FP), the Kenyan Ministry of Health introduced the comprehensive postnatal care guidelines that advocated for postpartum FP integration within antenatal care, postnatal care and infant immunization clinics at facility and community levels [21] to allow women to be counselled and offered postpartum FP options before delivery, at 6 weeks, 14 weeks, 6 months, and 9 months post-delivery. However, there are limited data on the factors associated with earlier postpartum FP uptake. We aimed to determine the timing of FP uptake within the first 9 months after delivery and the associated cofactors associated with earlier postpartum FP uptake in a large cohort of women in Western Kenya.

## Methods

### Study design and population

This study is a secondary data analysis of longitudinal data collected for the 'PrEP Implementation of Mothers in Antenatal care' (PrIMA) study, a cluster-randomized clinical trial that evaluated two models of pre-exposure prophylaxis (PrEP) delivery among pregnant women living in high HIV prevalence settings (risk-based versus universal delivery) – (NCT03070600) [22]. The study was conducted in 20 clinics in Siaya and Homa Bay counties in Kenya. From 15th January 2018–31st July 2019, women were recruited and enrolled into the study if they were pregnant, HIV-uninfected, age ≥ 15 years, tuberculosis negative, and planned to receive postnatal care at the facility of enrolment for at least one year postpartum. These women were then followed up using the Ministry of Health (MOH) maternal and child health (MCH) schedule for their routine antenatal care (ANC) and postpartum clinic visits until 9 months postpartum [23,24]. Each clinic enrolled a minimum of 200 pregnant HIV-uninfected women. For this analysis, we excluded women who did not attend any postnatal follow-up visits and women who had a miscarriage, stillbirth, or molar pregnancy in their current pregnancy.

**Data collection.** All study participants completed questionnaires at enrolment (during pregnancy), and in the postpartum period at 6 weeks, 14 weeks, 6 months, and 9 months. At enrolment, the questionnaires captured the women's sociodemographic information, partner characteristics, reproductive history, HIV risk, and social and mental health status including history of intimate partner violence (IPV). At each postnatal visit, the participants' postpartum sexual activity, pregnancy intention for the recently completed pregnancy and contraceptive methods use was recorded.

**Ethical considerations.** The study received ethical approval from the Kenyatta National Hospital-University of Nairobi (KNH-UoN) Ethics and Research Committee (P73/02/2017), and the University of Washington Institution Review Board (00000438). We obtained written informed consent from all participants prior to enrolment in either English, Kiswahili or Dholuo. Women 15–17 years of age included in this study were considered "emancipated minors" due to their pregnancy status allowing them to provide written informed consent without the presence of their guardians as stipulated in the Kenyan National Guidelines for HIV Testing and Counselling and the Kenya National Guidelines for Research and AIDS Vaccines/AIDS Vaccines [25–27].

**Inclusivity in global research.** Additional information regarding the ethical, cultural, and scientific considerations specific to inclusivity in global research is included in the Supporting Information (S1 Checklist "InclusivityInGlobalResearch").

**Definition of variables.** We defined modern FP methods as products or medical interventions that interfere with reproduction allowing women to have sexual intercourse at any desired time. These methods include tubal ligation, vasectomy, intrauterine contraceptive device (IUD), implants, injectables (Depo provera), oral contraceptive pills, and condoms (male or female) [2]. History of IPV was defined as having a score of ≥ 10 on the Hurt-Insult-Threaten-Scream (HITS) screening tool [28]. Having moderate-severe depressive symptoms was defined by having a score of ≥ 3 on the

Patient Health Questionnaire-2 (PHQ2) [29]. Social support was calculated using a multi-trait scaling analysis on the 18-item Medical Outcomes Study Social Support Survey (MOS-SSS) with a score >72 defined as high social support [30,31]. Since the parent study did not include a validated scale to measure pregnancy intention, pregnancies were characterized as intended, unintended or ambivalent based on questions that asked participants who had delivered what they felt about the last pregnancy. Participants were classified to have an intended pregnancy if they responded that they were trying to get pregnant; unintended pregnancy if they responded that they were trying to prevent a pregnancy and ambivalent if they responded that they did not care whether they became pregnant, it was God's will to become pregnant or not, or they did not know if they were trying to prevent a pregnancy. The timing of sexual resumption was based on the participants' self-report of sexual activity occurring prior to the specific postpartum visit. The risk of acquiring HIV was assessed using the rapid assessment screening tool (RAST) for PrEP eligibility scaled up by Kenya's HIV program [32]. The presence of any of the behavioral factors listed in the tool within the last 6 months was defined as a risk of acquiring HIV.

**Data analysis.** We utilized survival analysis methods to estimate the median time to uptake of modern postpartum FP methods [33]. The time to event was defined as the duration from the time the participant delivered to the earliest visit with a record of using postpartum FP on or before the 9-month visit. Participants were right-censored if they dropped out of the study at any follow-up visit postpartum and had not started modern postpartum FP, or if they had not started modern postpartum FP by the 9-month visit. Kaplan-Meier (KM) curves were used to estimate overall survival probability estimates while the Cox proportional hazard models using robust standard errors taking into account clustering by facility for the overall population were used to determine cofactors of earlier postpartum modern FP [34]. Potential cofactors for the cox proportional hazard models, including sociodemographic characteristics, pregnancy characteristics, and partnership dynamics, were selected *a priori* based on associations with uptake of postpartum FP noted in previous literature, or the research team's intuition informed by their clinical practice. We reported crude hazard ratios, and age-adjusted hazard ratios, with 95% confidence intervals and p-values. We conducted a sub-group analysis for participants who reported resuming sex by 9 months PP. We described postpartum FP methods that the participants took up overall and during study visits using counts and proportions. Data was analyzed using R 4.2.2.

## Results

### General characteristics

Overall, 4,447 pregnant women were enrolled, of whom 4,191 (94.2%) had a live birth and attended at least one postnatal visit during the follow-up period. We excluded 256 women who either experienced a miscarriage/stillbirth/molar pregnancy (n = 109) or did not attend any postnatal visit (n = 147).

Of 4,191 women included in the analysis, the median age was 24 years (interquartile range [IQR]: 21–28) with 17% of the women aged 15–19 years. Most women (88%) were in a steady relationship, were multigravida (75%), and 50% indicated that they intended to be pregnant. Eighty-nine percent of multigravida women had 4 or fewer living children with 49% of the women reporting that they had at least one male child. The median age of the youngest child at enrollment was 3 years (IQR: 2–5). Thirty-seven percent of women reported low social support, 8% reported history of IPV, 10% had moderate-to-severe depressive symptoms, and 13% were at high risk of acquiring HIV[26]. The median age of the male partners was 30 years (IQR: 26–35), with most (86%) women reporting that they were currently residing with their partners and 97% receiving financial support from their partners (Table 1).

### Uptake of postpartum family planning methods

Of 4,191 women, 45 (1%) attended ≥ 1 visit but were lost to follow-up before the 9-month visit, and 803 (19%) had not started any modern FP method by completion of the study at 9 months postpartum. Of the 803 women, < 1% used non-modern FP methods. At 6 weeks postpartum, 42% (1,759/4,191) of all women had resumed sex and 12% (492/4,191) took up postpartum FP. Approximately 79% (3,310/4,191) of all women had resumed sex by 14 weeks postpartum while

**Table 1. Baseline characteristics of postpartum women eligible for FP.**

| Characteristics | N | n (%) or median (IQR) |
|---|---|---|
| **Sociodemographic** | | |
| Age | 4189 | 24 (20.9 – 28.4) |
| Age groups | 4189 | |
| 15–19 years | | 708 (16.9%) |
| 20–24 years | | 1421 (33.9%) |
| more than 24 years | | 2060 (49.2%) |
| Completed primary school | 4095 | 3533 (86.3%) |
| High HIV risk by NASCOP* | 4191 | 525 (12.5%) |
| High HIV risk by Pintye et al** | 4191 | 1547 (36.9%) |
| Has regular employment | 4120 | 614 (14.9%) |
| Low social support | 4079 | 1498 (36.7%) |
| IPV (High HITS score ≥10)*** | 4166 | 325 (7.8%) |
| Moderate-to-severe depression (PHQ-2 score ≥3)**** | 3903 | 371 (9.5%) |
| **Pregnancy characteristics** | | |
| More than 4 ANC visits attended | 4034 | 2973 (73.7%) |
| Multigravida | 4169 | 3116 (74.7%) |
| Number of living children (excluding Primigravida) | 3116 | 3044 |
| No living child | | 168 (5.5%) |
| ≤4 children | | 2699 (88.7%) |
| >4 children | | 177 (5.8%) |
| Male child (excluding Primigravida) | 2833 | 1376 (48.6%) |
| History of miscarriage (excluding Primigravida) | 3113 | 433 (13.9%) |
| History of stillbirths (excluding Primigravida) | 3110 | 118 (3.8%) |
| Age of youngest child (excluding primigravida) | 2972 | |
| Below 2 years | | 603 (20.3%) |
| 2 – 4.9 years | | 1585 (53.3%) |
| 5 and above | | 784 (26.4%) |
| Pregnancy intention | 4162 | |
| Intended | | 2067 (49.7%) |
| Not intended | | 668 (16.1%) |
| Ambivalent | | 1427 (34.3%) |
| **Partner Characteristics (among women with primary partner at enrollment)** | | |
| With primary partner at enrolment | 4152 | 3824 (92.1%) |
| Partner age (years) | 3209 | 30 (26.0 – 35.0) |
| Partner HIV status at enrollment | 3810 | |
| Negative | | 2478 (65.0%) |
| Positive | | 166 (4.4%) |
| Unknown | | 1166 (30.6%) |
| Partner's completed primary school | 3684 | 3346 (90.8%) |
| Currently residing with partner | 3799 | 3258 (85.8%) |
| Partner provision of financial support | 3794 | 3692 (97.3%) |

* Risk assessment tool used by the Kenyan Ministry of Health to assess eligibility for PrEP.

** Objective risk assessment tool shown for identifying pregnant and postpartum women with good performance for predicting HIV risk (AUC 0.76 [95% CI: 0.67- 0.85]).

*** Evaluated using Hurt-Insult-Threaten-Scream (HITS) screening tool with those ≥10 being defined as experiencing IPV.

***** Assessed using the Patient Health Questionnaire-2 (PHQ2) with those with a score of ≥ 3 classified as having moderate-to-severe depressive symptoms.

only 38% (1,607/4,191) took up postpartum FP by this time, 88% (3,680/4,191) resumed sex versus 59% initiated post-partum FP (2,488/4,191) at 6 months postpartum and 91% (3,826/4,191) resumed sex versus 80% (3,343/4,191) initiated postpartum FP by 9 months postpartum (Fig 1). The median time to resumption of sexual activity was 8 weeks (IQR: 4–12 weeks) while the median time to uptake of postpartum FP was 24 weeks (IQR: 14–36 weeks) for the overall population and among women who resumed sex (p<0.001). Among the 3,826 women who resumed sex, 12% (475/3,826) took up at least 1 method of postpartum FP at 6 weeks, 41%(1,557/3,826) at 14 weeks, 63%(2,406/3,826) at 6 months and 83.6% (3,200/3,826)% at 9 months postpartum (Fig 2).

The most commonly used methods of modern postpartum FP were injectable (42%) and implants (40%). Injectables were the most common method of choice for women who took up postpartum FP at 6 weeks (45%), 14 weeks (47%) and 6 months (59%), and implants were the most common method of choice for women who first took up postpartum FP at 9 months (69%). Approximately 3.3% (138) of all women became pregnant during the follow-up period through 9-months postpartum for a pregnancy incidence of 7.18/100 person-years.

### Factors associated with family planning uptake

Earlier time to postpartum FP initiation was associated with older age (HR: 1.01 per increasing year, 95% CI: 1.00-1.01, p<0.001); and among women who had 4 children or less (aHR: 1.27, 95% CI: 1.05-1.53, p=0.013). Women without a partner at enrolment (aHR: 0.69, 95% CI: 0.58-0.82, p<0.001); not residing with their partners (aHR: 0.67, 95% CI: 0.62-0.72, p<0.001); not receiving financial support from their partner (aHR: 0.69, 95% CI: 0.59-0.79, p<0.001); who did not complete primary education (aHR: 0.86, 95% CI: 0.79-0.94, p<0.001); and who reported low social support (aHR: 0.88, 95% CI: 0.80-0.97, p=0.008) had later uptake of postpartum FP. Additionally, women who were primigravida (aHR: 0.71, 95% CI: 0.64-0.81, p<0.001); had a history of stillbirth (aHR: 0.75, 95% CI: 0.59-0.96, p=0.023); had a history of miscar-riage (aHR: 0.86, 95% CI:0.76-0.97, p=0.014); or whose youngest child at enrolment was below 2 years (aHR: 0.87, 95% CI: 0.77-0.98, p=0.025) had later postpartum FP uptake. Pregnancy intention was also associated with postpartum FP

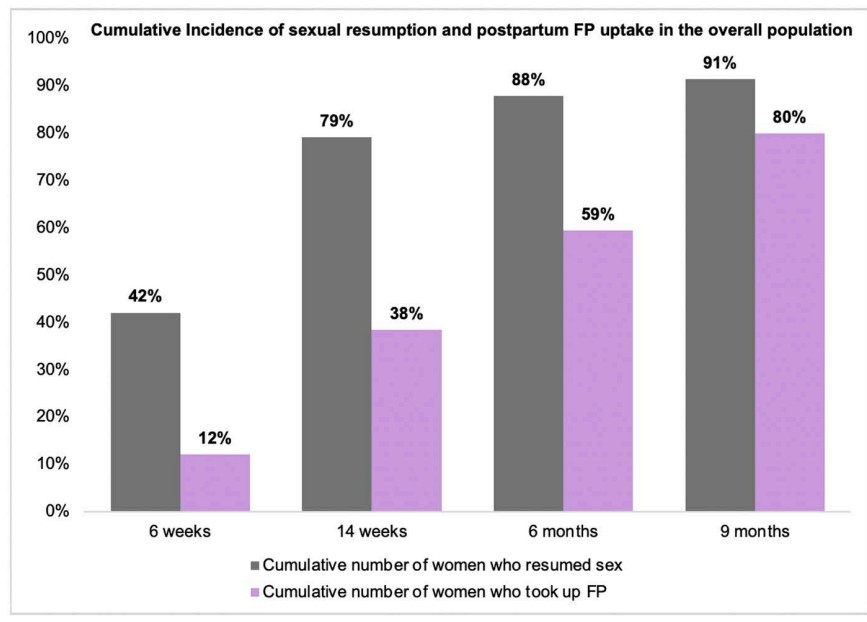

**Fig 1. Cumulative Incidence of sexual resumption and postpartum FP uptake in the overall population.**

**PLOS** **Global Public Health**

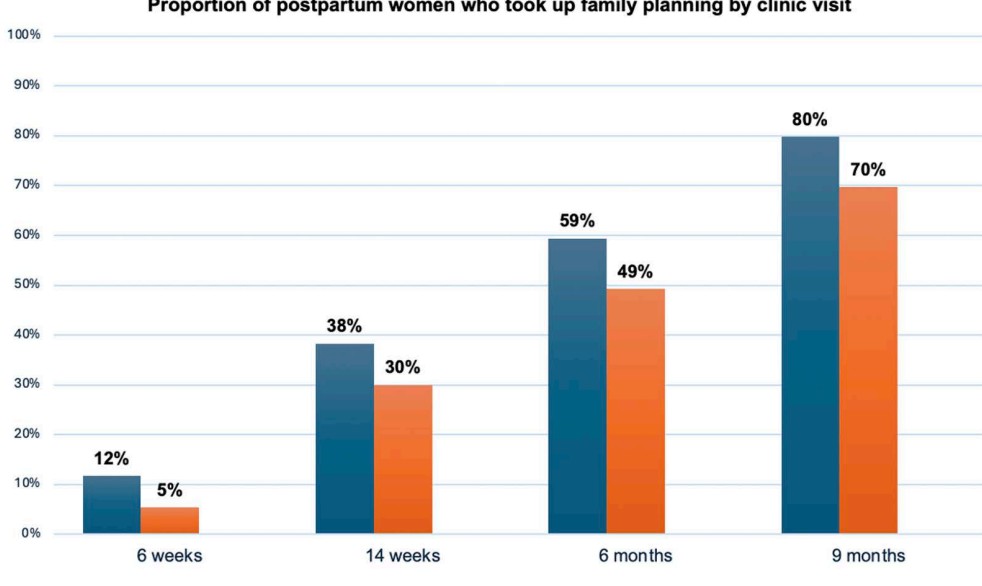

**Fig 2. Proportion of postpartum women who took up family planning by clinic visit.**

uptake. Women who were ambivalent (aHR: 0.79, 95% CI: 0.71-0.89, p < .0001) had later uptake postpartum FP compared to women who intended to become pregnant (Fig 3 and 4). These cofactors were similar in the overall population and among women who resumed sex (Table 2).

## Discussion

In this study conducted among postpartum women living in a high HIV prevalence region, approximately a fifth of the women did not start any modern FP method by 9-months postpartum. These findings are consistent with findings in several studies conducted in Kenya where the prevalence of postpartum FP use ranges from 47% - 78% [16,35,36]. Resumption of sex preceded uptake of postpartum FP by a median of 16 weeks. The most common method of modern postpartum FP used was injectables, though implants were popular among those initiating postpartum FP in later months. Later uptake of postpartum FP was observed among adolescent girls aged 15–19 years, women with lower levels of education, without a partner at that moment, not residing with their partners, with lower social support and no financial support from their partners. Additionally, later uptake of postpartum FP was observed among primigravid women, women who had a history of miscarriage/stillbirth, those whose youngest child at enrolment was below 2 years, and women who were ambivalent about their recent pregnancy.

Although most postpartum women want to delay subsequent pregnancies for 2–3 years [37], resumption of sex often precedes uptake of postpartum FP. Later postpartum FP uptake may be primarily driven by the assumption that breastfeeding postpones ovulation until the resumption of menses. While breastfeeding has been shown to delay the return to fertility among postpartum women [38], it is highly dependent on the intensity of the child's suckling and the duration and frequency of breastfeeding per day [39]. Additionally, lactating women may ovulate before the resumption of menses [40]. This places postpartum women at risk of mistimed and unintended pregnancy, and may explain the high rate of short inter-pregnancy intervals in Kenya, and predispose women to maternal and neonatal complications [41,42]. There is a need to close the gap in the provision and uptake of safe and effective modern postpartum FP methods soon after

**Kaplan-Meier survival plots for different cofactors of earlier uptake of modern postpartum family planning methods in the overall population**

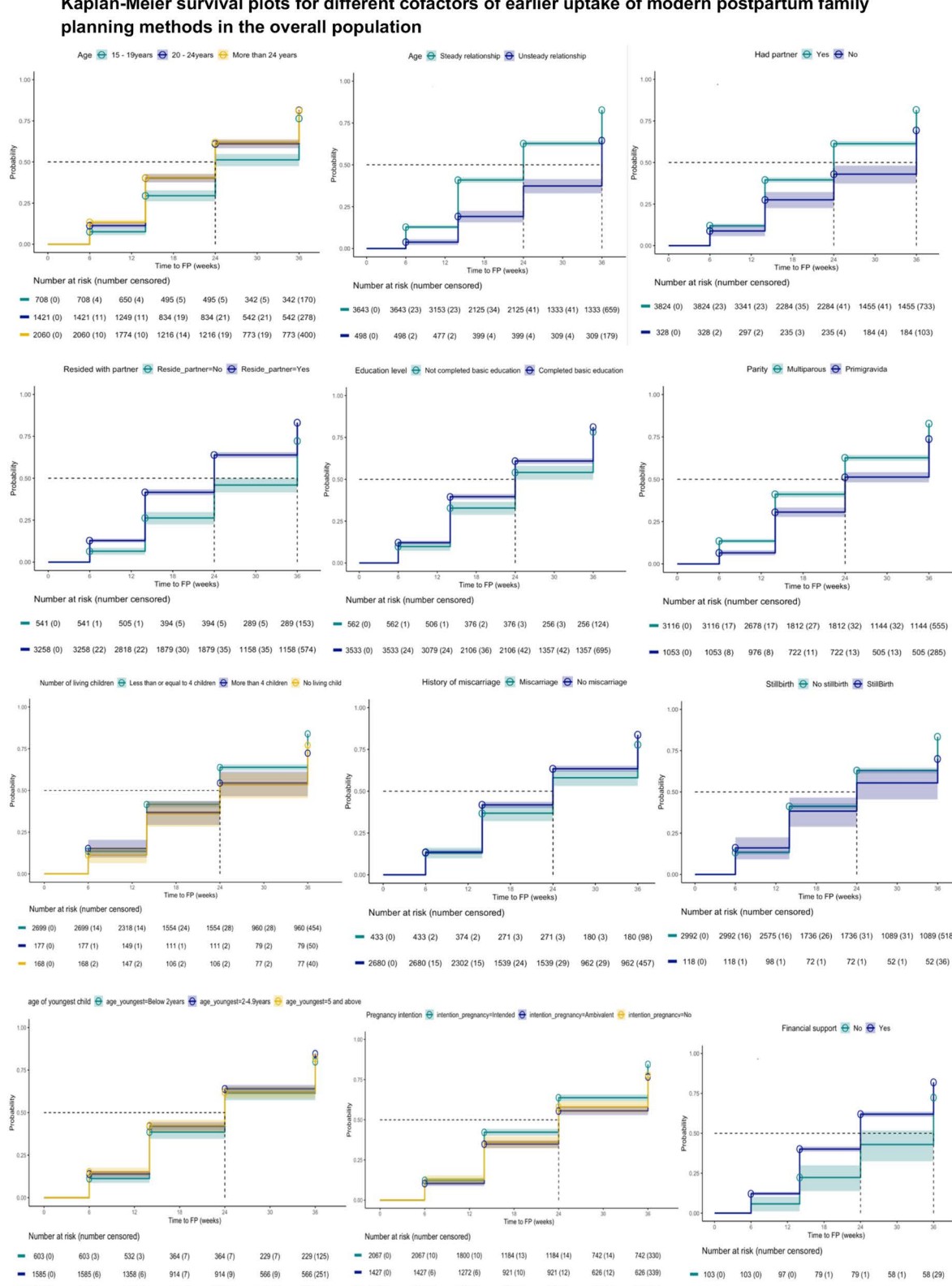

**Fig 3. Kaplan-Meier survival plots for different cofactors of earlier uptake of modern postpartum family planning methods in the overall population.**

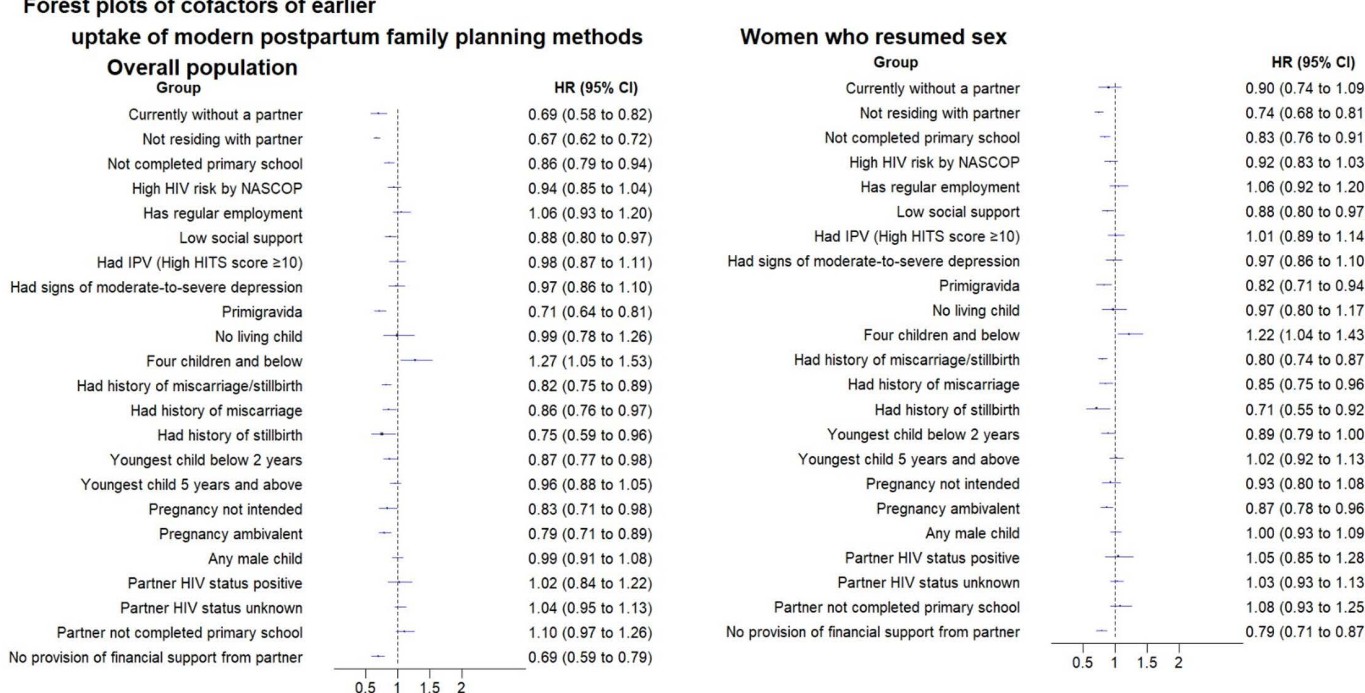

**Fig 4. Forest plots of cofactors of earlier uptake of modern postpartum family planning methods.**

delivery, and before resumption of sexual activity, reducing the unmet need for FP in this population [43]. Development of tailored, context-specific, non-formal education and outreach programs targeting antenatal and postpartum women may prompt earlier initiation of postpartum FP [44–46]. These programs should emphasize on the importance of postpartum FP, provide information on the available modern FP methods and specifically address the existing myths and misconceptions.

Uptake of long-acting reversible contraceptives (IUDs and implants) among postpartum women remains low even though the correct and consistent use of short-term methods (injectables, oral contraceptive pills and condoms) is a challenge [47]. Consistent with other studies [47], our analysis found that injectables were the most common method of choice for postpartum women. Short-term hormonal methods predispose postpartum women to FP discontinuation and increased contraceptive failure rates [48,49] as they are user-dependent and require consistent reminders resulting in unintended/mistimed pregnancies [50,51]. In contrast to non-pregnant women, postpartum women have more opportunities to receive LARC following their regular interaction with the healthcare system during their antenatal visit, postnatal and well-baby visits. However, an important gap exists in the provision of long-acting reversible contraceptives (LARC) methods. Strategies to overcome existing community-, facility- and individual-level barriers to uptake of LARC including financial subsidies for LARCs and the incorporation of counseling on LARC during antenatal care, immediately after delivery and during subsequent postnatal visits is critical [52,53].

This study highlighted key individual and interpersonal factors influencing the timing of postpartum FP uptake. At the individual level, attitudes towards pregnancy and perceived need for FP are an important contributor to uptake of modern postpartum FP [54,55]. Similar to other studies [56], our study demonstrated that women who have unresolved/contradictory feelings about their pregnancies are likely to delay modern postpartum FP [57]. Women who are ambivalent about pregnancy have unique needs and concerns about their fertility. Further studies on the family planning needs and desires of

Table 2. Correlates of early time to postpartum FP initiation.

| Variable | Overall population Unadjusted HR(95% CI) | p-value | Adjusted for age HR (95% CI) | p-value | Those who resumed sex Unadjusted HR (95% CI) | p-value | Adjusted for age HR (95% CI) | p-value |
|---|---|---|---|---|---|---|---|---|
| **Actual Age** | 1.01 (1.00 – 1.01) | <0.001 | | | 1.00 (1.00 – 1.01) | 0.570 | | |
| Age groups | | | | | | | | |
| **15-19 years** | 0.80 (0.74 – 0.87) | <0.001 | | | 0.89 (0.82 – 0.96) | 0.003 | | |
| 20–24 years | 0.99 (0.92 – 1.07) | 0.818 | | | 1.00 (0.92 – 1.09) | 0.958 | | |
| More than 24 years | ref | | | | ref | | | |
| Currently with partner | | | | | | | | |
| **Currently without a partner** | 0.67 (0.57 – 0.80) | <0.001 | 0.69 (0.58 – 0.82) | <0.001 | 0.90 (0.75 – 1.08) | 0.251 | 0.90 (0.74 – 1.09) | 0.287 |
| Currently with a partner | ref | | ref | | | | | |
| Residing with partner | | | | | | | | |
| **Not residing with partner** | 0.67 (0.62 – 0.73) | <0.001 | 0.67 (0.62 – 0.72) | <0.001 | 0.75 (0.69 – 0.82) | <0.001 | 0.74 (0.68 – 0.81) | <0.001 |
| Residing with partner | ref | | ref | | | | | |
| Participant education level | | | | | | | | |
| **Not completed primary school** | 0.87 (0.80 – 0.96) | 0.004 | 0.86 (0.79 – 0.94) | <0.001 | 0.84 (0.77 – 0.92) | <0.001 | 0.83 (0.76 – 0.91) | <0.001 |
| Completed primary school | ref | | ref | | | | | |
| HIV risk by NASCOP | | | | | | | | |
| High HIV risk by NASCOP | 0.94 (0.84 – 1.05) | 0.248 | 0.94 (0.85 – 1.04) | 0.201 | 0.92 (0.83 – 1.02) | 0.133 | 0.92 (0.83 – 1.03) | 0.138 |
| Low HIV risk by NASCOP | ref | | ref | | | | | |
| Participant employment | | | | | | | | |
| Has regular employment | 1.08 (0.95 – 1.23) | 0.222 | 1.06 (0.93 – 1.20) | 0.400 | 1.05 (0.93 – 1.20) | 0.380 | 1.06 (0.92 – 1.20) | 0.424 |
| Does not have regular employment | ref | | ref | | | | | |
| Low social support* | | | | | | | | |
| **Yes** | 0.88 (0.80 – 0.97) | 0.009 | 0.88 (0.80 – 0.97) | 0.008 | 0.88 (0.80 – 0.98) | 0.014 | 0.88 (0.80 – 0.97) | 0.012 |
| No | ref | | ref | | | | | |
| History of IPV | | | | | | | | |
| Had IPV (High HITS score ≥10)*** | 1.00 (0.89 – 1.14) | 0.957 | 0.98 (0.87 – 1.11) | 0.804 | 1.00 (0.89 – 1.14) | 0.889 | 1.01 (0.89 – 1.14) | 0.924 |
| No IPV | ref | | ref | | | | | |
| Signs of moderate-to-severe depression (PHQ-2 score ≥3)**** | | | | | | | | |
| Had signs of moderate-to-severe depression | 0.98 (0.86 – 1.10) | 0.695 | 0.97 (0.86 – 1.10) | 0.654 | 0.97 (0.86 – 1.10) | 0.665 | 0.97 (0.86 – 1.10) | 0.663 |
| No signs of moderate-to-severe depression | ref | | ref | | | | | |
| **Pregnancy characteristics** | | | | | | | | |
| Parity | | | | | | | | |
| **Primigravida** | 0.74 (0.67 – 0.81) | <0.001 | 0.71 (0.64 – 0.81) | <0.001 | 0.84 (0.74 – 0.95) | 0.003 | 0.82 (0.71 – 0.94) | 0.004 |
| Multigravida | ref | | ref | | | | | |
| Number of living children (excluding Primigravida) | | | | | | | | |
| No living child | 1.07 (0.86 – 1.32) | 0.562 | 0.99 (0.78 – 1.26) | 0.933 | 1.04 (0.86 – 1.25) | 0.696 | 0.97 (0.80 – 1.17) | 0.747 |

(Continued)

Table 2. (Continued)

| Variable | Unadjusted — Overall population HR(95% CI) | p-value | Adjusted for age HR (95% CI) | p-value | Unadjusted — Those who resumed sex HR (95% CI) | p-value | Adjusted for age HR (95% CI) | p-value |
|---|---|---|---|---|---|---|---|---|
| ≤4 children | 1.33 (1.15 – 1.55) | <0.001 | 1.27 (1.05 – 1.53) | 0.013 | 1.28 (1.13 – 1.45) | <0.001 | 1.22 (1.04 – 1.43) | 0.015 |
| >4 children | ref | | | | | | | |
| History of miscarriage/stillbirth | | | | | | | | |
| Had history of miscarriage/stillbirth | 0.81 (0.74 – 0.88) | <0.001 | 0.82 (0.75 – 0.89) | <0.001 | 0.79 (0.73 – 0.86) | <0.001 | 0.80 (0.74 – 0.87) | <0.001 |
| No history of miscarriage | ref | | | | | | | |
| History of miscarriage | | | | | | | | |
| Had history of miscarriage | 0.85 (0.75 – 0.96) | 0.009 | 0.86 (0.76 – 0.97) | 0.014 | 0.84 (0.74 – 0.96) | 0.008 | 0.85 (0.75 – 0.96) | 0.012 |
| No history of miscarriage | ref | | | | | | | |
| History of stillbirths | | | | | | | | |
| Had history of stillbirth | 0.74 (0.58 – 0.95) | 0.017 | 0.75 (0.59 – 0.96) | 0.023 | 0.70 (0.55 – 0.91) | 0.007 | 0.71 (0.55 – 0.92) | 0.009 |
| No history of stillbirth | ref | | | | | | | |
| Age of youngest child | | | | | | | | |
| Below 2 years | 0.89 (0.79 – 0.99) | 0.037 | 0.87 (0.77 – 0.98) | 0.025 | 0.90 (0.81 – 1.01) | 0.062 | 0.89 (0.79 – 1.00) | 0.043 |
| 2 – 4.9 years | ref | | | | | | | |
| 5 and above | 0.95 (0.87 – 1.03) | 0.226 | 0.96 (0.88 – 1.05) | 0.354 | 0.99 (0.90 – 1.10) | 0.921 | 1.02 (0.92 – 1.13) | 0.694 |
| Pregnancy intention | | | | | | | | |
| Yes | ref | | ref | | ref | | | |
| Not intended | 0.83 (0.70 – 0.98) | <0.001 | 0.83 (0.71 – 0.98) | 0.027 | 0.93 (0.80 – 1.08) | 0.361 | 0.93 (0.80 – 1.08) | 0.363 |
| Ambivalent | 0.79 (0.71 – 0.89) | 0.027 | 0.79 (0.71 – 0.89) | <0.001 | 0.87 (0.78 – 0.96) | 0.008 | 0.87 (0.78 – 0.96) | 0.008 |
| Any male child | 0.99 (0.91 – 1.10) | 0.811 | 0.99 (0.91 – 1.08) | 0.868 | 1.00 (0.92 – 1.09) | 0.976 | 1.00 (0.93 – 1.09) | 0.939 |
| **Partner Characteristics** | | | | | | | | |
| Partner HIV status at enrollment | | | | | | | | |
| Positive | 1.02 (0.85 – 1.23) | 0.804 | 1.02 (0.84 – 1.22) | 0.875 | 1.04 (0.85 – 1.28) | 0.686 | 1.05 (0.85 – 1.28) | 0.670 |
| Unknown | 1.04 (0.95 – 1.14) | 0.406 | 1.04 (0.95 – 1.13) | 0.427 | 1.03 (0.94 – 1.13) | 0.565 | 1.03 (0.93 – 1.13) | 0.573 |
| Negative | ref | | | | ref | | Ref | |
| Partner completed primary school | | | | | | | | |
| Not completed primary school | 1.11 (0.97 – 1.27) | 0.132 | 1.10 (0.97 – 1.26) | 0.149 | 1.08 (0.93 – 1.25) | 0.295 | 1.08 (0.93 – 1.25) | 0.304 |
| Completed primary school | Ref | | | | | | | |
| Partner provision of financial support | | | | | | | | |
| No provision of financial support | 0.68 (0.59 – 0.79) | <0.001 | 0.69 (0.59 – 0.79) | <0.001 | 0.79 (0.71 – 0.88) | <0.001 | 0.79 (0.71 – 0.87) | <0.001 |

ambivalent women are warranted. Additionally, postpartum women in unsteady relationships, without partners and those living away from their partners, may perceive themselves to have a lower risk of pregnancy potentially affecting their motivation to take up postpartum FP. There is a need to design specialized postpartum FP delivery strategies and counselling programs that provide targeted options for postpartum women with diverse pregnancy intentions and ongoing interpersonal factors to meet their evolving FP needs, increase early postpartum FP demand and FP self-efficacy. These programs should include initiatives to assess the individual FP needs of postpartum women, foster partner involvement to promote shared responsibility for postpartum FP use and encourage community-based social support for uptake and use of postpartum FP.

Adolescent pregnancy is a major public health problem with an accompanying high rate of maternal and child morbidities and mortalities [58]. In Kenya, approximately 15% of adolescent girls aged 15–19 years have ever been pregnant. Homa Bay [59] and Siaya counties are among the top 10 counties with high rates of adolescent pregnancies (23% and 21% respectively) [6]. Although national efforts are focused on reducing the adolescent pregnancy rates, very slow progress has been made. In a region where the age of sexual debut among women is 16 years [60] and the prevalence of HIV is high [61], provision of dual methods of FP and HIV prevention is critical to prevent unwanted pregnancies and new HIV infections [62]. Unfortunately, uptake of FP methods is low in this population [63–65]. This may be primarily driven by stigma, myths and misconceptions, and service provider reluctance to offer FP in this population without parental consent [63,64] as is currently the requirement stipulated in the Children's Act and the National Reproductive Health Policy, Kenya [66,67]. Considering that adolescent girls are predisposed to having multiple, risky and age-disparate sexual partners [68], country-level advocacy to ease adolescents' access to FP methods and scale-up of tailored sexual and reproductive health programs, including information on FP in schools and in the healthcare facilities is essential [69].

Our study had several strengths. First, this study involved a large cohort of postpartum women from multiple clinics. Moreover, the study recruited the women while pregnant and observed them during the postpartum period. We were therefore able to ascertain the time when resumption of sex and uptake of postpartum FP occurred prospectively. This study had some limitations. Although the study visits mirrored the required postnatal and infant immunization visits, we excluded women who did not attend any postnatal follow-up visits to determine the median time to postpartum resumption of sexual activities and uptake of modern postpartum FP method. By doing so, we may have introduced selection bias by capturing women who have higher health-seeking behaviors potentially affecting the generalizability of the findings. Additionally, excluding women who experienced miscarriages, stillbirths, or molar pregnancies may limit the understanding of the needs and impact of adverse pregnancy outcomes on uptake of postpartum FP. Moreover, this study only included women who were not living with HIV and there may be differences in postpartum FP uptake between these women and women living with HIV. Finally, the pregnancy intentions of these women were not measured using a validated scale such as the London Measure of Unplanned Pregnancy (LMUP).

## Conclusion

Understanding the timing of postpartum FP uptake and the cofactors that drive earlier uptake is critical to inform and strengthen the programming strategies for the existing postpartum family planning interventions. Delays in postpartum FP initiation are driven by individual, interpersonal, and obstetric factors that should be taken into consideration during the development of postpartum FP interventions. Strategies to increase uptake of long-acting reversible contraceptives, and understand the evolving sexual and reproductive needs of postpartum women are urgently needed to reduce the unmet need of FP in this population. Further research is needed on different strategies for delivery of FP to postpartum women to overcome delays in FP initiation, taking into account their individual, interpersonal, and obstetric needs and perspectives.

## Supporting information

**S1 Checklist. Inclusivity in global research.**
(DOCX)

## Acknowledgments

We would like to thank the participants for their contributions to the study and the PrIMA study team for the data collection. We also thank the Kenyan Ministry of Health, Homa Bay and Siaya counties for their collaboration.

## Author contributions

**Conceptualization:** Jared M. Baeten, John Kinuthia, Grace John-Stewart.

**Data curation:** Grace John-Stewart.

**Formal analysis:** Damaris Kimonge, Mary M. Marwa, Salphine Watoyi.

**Funding acquisition:** Jared M. Baeten, Grace John-Stewart.

**Investigation:** Jared M. Baeten, John Kinuthia, Grace John-Stewart.

**Methodology:** Jillian Pintye, Jared M. Baeten, John Kinuthia, Grace John-Stewart.

**Project administration:** Julia C. Dettinger, Felix Abuna, Laurén Gómez, Emmaculate Nzove.

**Supervision:** Julia C. Dettinger, Felix Abuna, Ben Odhiambo, Laurén Gómez, Anjuli D. Wagner, Jillian Pintye, John Kinuthia, Grace John-Stewart, Cyrus Mugo.

**Validation:** Anjuli D. Wagner, Mary M. Marwa, Jillian Pintye, John Kinuthia, Grace John-Stewart, Cyrus Mugo.

**Writing – original draft:** Nancy Mwongeli Ngumbau.

**Writing – review & editing:** Julia C. Dettinger, Felix Abuna, Ben Odhiambo, Laurén Gómez, Anjuli D. Wagner, Mary M. Marwa, Salphine Watoyi, Emmaculate Nzove, Jillian Pintye, Jared M. Baeten, John Kinuthia, Grace John-Stewart, Cyrus Mugo.

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
