## [Decision Letter · Decision Letter 0]

11 Sep 2024

PGPH-D-24-01009

Cofactors of earlier uptake of modern postpartum family planning methods in Kenya

Dear Dr. Ngumbau,

Thank you for submitting your manuscript to PLOS Global Public Health. After careful consideration, we feel that it has merit but does not fully meet PLOS Global Public Health’s publication criteria as it currently stands. Therefore, we invite you to submit a revised version of the manuscript that addresses the points raised during the review process.

Comments from two reviewers are copied below (Reviewer 1) and in the attached document (Reviewer 2's main points are in the attachment). Please read through all comments, including those in the attachment, and be sure to respond to all of them in your rebuttal letter.

We look forward to receiving your revised manuscript.

Kind regards,

Mark C. Wheldon, Ph.D.

Academic Editor

Journal Requirements:

Additional Editor Comments (if provided):

Please note that Reviewer 2 has submitted comments in an attachment. Please let us know if you cannot access the attachment with Reviewer 2's comments.

Reviewers' comments:

Reviewer's Responses to Questions

**Comments to the Author**

1. Does this manuscript meet PLOS Global Public Health’s publication criteria?

Reviewer #1: Yes

Reviewer #2: Yes

2. Has the statistical analysis been performed appropriately and rigorously?

Reviewer #1: Yes

Reviewer #2: Yes

3. Have the authors made all data underlying the findings in their manuscript fully available (please refer to the Data Availability Statement at the start of the manuscript PDF file)?

Reviewer #1: Yes

Reviewer #2: Yes

4. Is the manuscript presented in an intelligible fashion and written in standard English?

Reviewer #1: Yes

Reviewer #2: Yes

Reviewer #1: This is an important topic and a well conducted study.

Suggested Edits:

1. Abstract

Background: Consider adding ‘in high HIV prevalence settings’ for accuracy and representation of your study population.

For example: “There are limited data on uptake of postpartum family planning (FP) in resource-limited settings, particularly in high HIV prevalence settings. We assessed the timing of postpartum FP initiation and the cofactors of earlier uptake of modern postpartum FP in Kenya”

Conclusion: For precision, consider highlighting the contribution of your study to the research.

For example: Resumption of sex occurred earlier than uptake of postpartum FP. Our findings underscore the importance of addressing individual, interpersonal, and obstetric factors to tailor postpartum FP counseling and interventions, particularly in high HIV prevalence settings.

2. Introduction

The introduction jumps between global and Kenya-specific statistics. Reorganize to present a clearer narrative and improve the flow:

a. Global burden of unmet need for FP, with a focus on sub-Saharan Africa.

b. FP situation in Kenya (national and regional variations).

c. Importance of postpartum FP and the knowledge gaps this study addresses.

3. Methods

Though it is mentioned, consider clearly stating that this is a secondary analysis of data from a larger study in your study design.

While the definition of ‘Modern FP’ is provided, consider adding a brief explanation of why this distinction is important (e.g., effectiveness, user dependence).

Acknowledge that a validated scale was not used to determine ‘Pregnancy Intention’ and explain the rationale for the chosen method, as mentioned in the ‘Discussion’ section.

In your data analysis, specify the time origin for the survival analysis (date of delivery). Also, briefly describe the process of selecting covariates for the Cox models.

4. Results

Consider a more informative caption for figure 1, such as "Cumulative Incidence of Sexual Resumption and Postpartum FP Uptake."

I wonder if a forest plot would complement figure 3 to present hazard ratios and confidence intervals more effectively.

5. Discussion

Good summary of key findings; Good association with existing literature; exhaustive strengths and limitations (large sample size, prospective design, and use of appropriate statistical methods)

Consider adding a specific and actionable recommendation for interventions and policies (Is it generalizable? Consider HIV +ve population) such as counseling on the risks of pregnancy during the postpartum period and the benefits of long-acting reversible contraceptives (Or peer support groups, financial incentives, and male involvement programs). Owing to the study’s significance, a call for further research to understand the reasons behind the observed associations is appropriate.

Reviewer #2: The findings highlight critical barriers to postpartum family planning (FP) uptake among vulnerable groups, particularly adolescent girls and women with lower education levels. Targeted educational programs are essential to address misconceptions and promote awareness of available FP methods. Additionally, enhancing partner involvement in family planning decisions can create a supportive environment that encourages uptake.

Social support networks are crucial for women without partners, as they can provide emotional and informational resources. Financial assistance programs should be established to help women access contraceptive methods, particularly those who lack financial support from their partners.

Moreover, addressing the unique needs of primigravid women and those with a history of adverse pregnancy outcomes through specialized counseling can significantly improve their engagement with FP services. Integrating FP services into routine maternal health care visits will ensure that discussions about family planning occur consistently, particularly for women with young children.

Regarding research ethics, it is vital to ensure that the study adheres to ethical standards, including informed consent and confidentiality. Concerns about dual publication should also be addressed; any overlapping content with previous studies should be clearly cited to avoid issues of academic integrity. Transparency in methodology and findings is essential for maintaining trust in the research process. Overall, these recommendations and considerations will strengthen the study's impact and contribute to improved maternal health outcomes in Kenya.

**Do you want your identity to be public for this peer review?** For information about this choice, including consent withdrawal, please see our Privacy Policy

Reviewer #1: **Yes: ** Lydia Wangui Ngigi

Reviewer #2: **Yes: ** LYDIA CHERUTO PKAREMBA

---

## [Editor Report · Decision Letter 1]

20 Dec 2024

PGPH-D-24-01009R1

Cofactors of earlier uptake of modern postpartum family planning methods in Kenya

Dear Dr. Ngumbau,

Thank you for submitting your manuscript to PLOS Global Public Health. After careful consideration, we feel that it has merit but does not fully meet PLOS Global Public Health’s publication criteria as it currently stands. Therefore, we invite you to submit a revised version of the manuscript that addresses the points raised during the review process.

In your letter and revised manuscript with track changes I can only see responses to the requests from the editorial office in sections "Competing interests" and "Funding". However, there were specific comments and requests from two reviewers, pasted inline in my decision letter sent 11th September. There was also an attached document with one of the sets of comments. Please refer back to that decision letter sent 11th September and prepare a point-by-point response to the two reviewers' comments and make appropriate revisions to the manuscript. Further, please describe any changes to the manuscript or submit a version with track changes.

If you cannot see the comments from the two reviewers in the original decision email sent 11th September please write back to get these comments re-sent.

Regards,

Mark Wheldon, handling editor.

We look forward to receiving your revised manuscript.

Kind regards,

Mark C. Wheldon, Ph.D.

Academic Editor
---

## [Decision Letter · Decision Letter 2]

3 Feb 2025

PGPH-D-24-01009R2

Cofactors of earlier uptake of modern postpartum family planning methods in Kenya

Dear Dr. Ngumbau,

Thank you for submitting your manuscript to PLOS Global Public Health. There is one small outstanding request (see below). Therefore, we invite you to submit a revised version of the manuscript that addresses the points raised during the review process.

Additional Editor Comments (if provided):

Thank you for revising the manuscript as requested. There is one small outstanding request from a reviewer that I did not see in the revision:

Results: Consider a more informative caption for figure 1, such as "Cumulative Incidence of Sexual Resumption and Postpartum FP Uptake."

Thank you for this comment. We have edited the caption for figure 1 to read:

“Cumulative incidence of sexual resumption and postpartum FP uptake in the overall population”

I did not see this edit in the revised version. If you could make this change it will be ready for acceptance.

We look forward to receiving your revised manuscript.

Kind regards,

Mark C. Wheldon, Ph.D.

Academic Editor

Journal Requirements:

Reviewers' comments:

Reviewer's Responses to Questions

**Comments to the Author**

Reviewer #2: All comments have been addressed

publication criteria?

Reviewer #2: Yes

3. Has the statistical analysis been performed appropriately and rigorously?

Reviewer #2: Yes

4. Have the authors made all data underlying the findings in their manuscript fully available (please refer to the Data Availability Statement at the start of the manuscript PDF file)?

Reviewer #2: Yes

5. Is the manuscript presented in an intelligible fashion and written in standard English?

Reviewer #2: Yes

Reviewer #2: The author discusses the history of contraceptive intake in Kenya, focusing on programs initiated in 1967. However, it is essential to highlight the evolution of these initiatives up to 2024, particularly current programs such as Universal Health Coverage (UHC). A comprehensive overview should include at least four critical programs that illustrate this progression. This synchronization from 1967 to 2024 showcases how Kenya has continuously adapted its family planning programs to meet the evolving needs of its population. Some programs include

1. Family Planning Program (1967): This program marked the beginning of organized contraceptive services in Kenya, establishing a foundation for future initiatives.

2. Reproductive Health Strategy (2009): This strategy expanded access to family planning services, emphasizing reproductive health education and the importance of informed choices.

3. Kenya's Universal Health Coverage (2018): As part of the Big Four Agenda, UHC aims to provide equitable access to essential health services, including family planning, thereby addressing the needs of all Kenyans.

4. Adolescent Sexual and Reproductive Health (ASRH) Program: Recently implemented, this program focuses on providing tailored education and outreach to adolescents, especially girls and women with lower levels of education, ensuring they receive appropriate information and services.

Additionally, the introduction of targeted education and outreach programs for adolescent girls and undereducated women was well addressed. The author also effectively commented on the importance of strengthening partner involvement by implementing initiatives that encourage shared decision-making in family planning discussions.

NB. Don’t forget to properly cite the programs

**Do you want your identity to be public for this peer review?** For information about this choice, including consent withdrawal, please see our Privacy Policy

Reviewer #2: **Yes: ** DR. LYDIA CHERUTO PKAREMBA

---

## [Editor Report · Decision Letter 3]

10 Feb 2025

Cofactors of earlier uptake of modern postpartum family planning methods in Kenya

PGPH-D-24-01009R3

Dear Dr Ngumbau,

We are pleased to inform you that your manuscript 'Cofactors of earlier uptake of modern postpartum family planning methods in Kenya' has been provisionally accepted for publication in PLOS Global Public Health.

Best regards,

Mark C. Wheldon, Ph.D.

Academic Editor